# The Relationship between Health Perception and Health Predictors among the Elderly across European Countries

**DOI:** 10.3390/ijerph18084053

**Published:** 2021-04-12

**Authors:** Ana F. Silva, Jose Mª Cancela, Irimia Mollinedo, Miguel Camões, Pedro Bezerra

**Affiliations:** 1Escola Superior Desporto e Lazer, Instituto Politécnico de Viana do Castelo, Rua Escola Industrial e Comercial de Nun’Álvares, 4900-347 Viana do Castelo, Portugal; joaocamoes@esdl.ipvc.pt (M.C.); pbezerra@esdl.ipvc.pt (P.B.); 2N2i, Polytechnic Institute of Maia, 4475-690 Maia, Portugal; 3The Research Centre in Sports Sciences, Health Sciences and Human Development (CIDESD), 5001-801 Vila Real, Portugal; 4HealthyFit Research Group, Institute of Health Research Galicia Sur (IISGS), Hospital University Complex of Pontevedra (CHOP), SERGAS, University of Vigo, 36310 Pontevedra, Spain; chemacc@uvigo.es (J.M.C.); imollinedo@uvigo.es (I.M.)

**Keywords:** elderly, strength, quality of life, agility, sex, country

## Abstract

This study aimed to investigate the relationship between health perception and health predictors among the elderly. In this study, 376 older adults from four different countries (Hungary, *n* = 86; Italy, *n* = 133; Portugal, *n* = 95; and Spain, *n* = 62) were analyzed. All subjects completed the EQ-5D-5L to assess their quality-adjusted life years and were assessed in handgrip (HG) and in Timed Up and Go (TUG) tests. A three-way MANOVA was conducted to analyze the groups based on their age, sex, and country. The interaction effects in all included variables were also considered. The Bonferroni test was also executed as a post hoc test. Any interaction results were noticed. Regarding age, lower perceived quality of life scores and higher TUG results were registered in the oldest group, and greater values of left and right HG results were registered in the second-oldest group. Males showed greater left and right HG values than women. Spain showed lower perceived quality of life scores. Portugal and Italy showed greater HG left values, while Portugal had better HG right values. Hungary produced the greatest TUG scores. Quality of life is dependent on the subject’s age and physical fitness, as increasing age was associated with decreased values of HG and TUG. Only strength was different between sexes.

## 1. Introduction

Human life expectancy has increased dramatically over the last 100 years. In fact, the European Union is already the world’s oldest region [1], with an aging rate of 94.1% in 2001, which further increased in 2017 to 125.8% [2,3]. That clear increase in life expectancy has led to a growing interest and need to improve the quality of life (QoL) of the elderly. This multifaceted and complex concept reflects the interaction of objective, subjective, macro, micro, positive, and negative influences in life [4]. Nevertheless, males and females report different health perceptions and, thus, QoL. For instance, elderly women had 25% more complaints of bad or very poor QoL when compared to elderly men [5]. Such sex-based differences were also observed with the finding that older-aged men are more susceptible to environmental factors than women [6], and higher levels of physical fitness were found to better contribute to a higher assessment of the QoL and health of men than women. Indeed, correlations of 0.75 and 0.90 between physical fitness and QoL were found (depending on whether a participant was living in a nursing home or in the University of the Third Age) [7], indicating the important role of physical fitness in self-assessed QoL.

Health-related QoL is clearly influenced by several endogenous and exogenous parameters [8,9]. Considering the endogenous parameters, physical well-being, or physical fitness, is undoubtedly one of the most notable [10]. For instance, the ability of balance—an ability directly involved in most activities of daily living—decreases with age, and when associated with declines in other physical abilities, such as strength, it can increase an individual’s exposure to falls, disability, cognitive impairments, depression, and even mortality [11,12,13]. The loss of muscle strength, which enhances the risk of fatal falls and the risk of fractures, is especially strongly related to reduced health-related QoL [14,15]. Once again, differences were observed between sexes, with a significantly higher occurrence of falls in females than males [16,17]. In fact, those decreases are caused by genetic factors [18], but also by lifetime physical activity [19,20] and one’s occupational position [21].

Regarding the exogenous parameters, socioeconomic status has been indicated as the main influencing factor to promote the highest health frailty scores. In fact, it was found that cardiovascular disease and mortality varied between high-income, middle-income, and low-income countries, with higher risks to the poorest countries [22]. Nevertheless, socioeconomic status is a multidimensional construct, related to both adequacy of financial resources and educational attainment [23]. The latter was found to be the most consistent indicator associated with socioeconomic status (more than wealth), with people with low education levels in low- and middle-income countries showing a markedly higher risk, for instance, of major cardiovascular events compared with those with higher levels of education [22].

It is well-known that muscle strength is considered vital for performing activities of daily living [12]. Since hand-grip strength (HG) is strongly related to lower-extremity muscle strength [24], as well as overall body strength [25], it has been referred to as an indicator of muscle strength for the entire body [12,26]. Numerous prospective studies have described the association between HG and declining health among the elderly, predominantly describing its association with functional disability [27,28] and mortality [29,30]. In fact, it has been shown that HG decreases significantly as age increases among men and women [31]. Elderly people with reduced HG tend to have a poor health-related QoL when compared to those with normal values [32]. An investigation of middle-aged and older adults who were followed for about seven years found that men and women in the highest HG quintile at baseline had a 32% and 25% lower risk, respectively, for all-cause mortality compared to those in the lowest HG quintile [33]. On the other hand, a recent review [34] assessed the effect of resistance training on physical functioning in subjects over 60 years old. Performing high-intensity strength training three times a week significantly improved muscle strength and was associated with improved physical abilities [28].

Lower extremity performance is associated with strength, but also with balance, mobility, and fall risk among the elderly and people with pathologies (i.e., Parkinson’s disease, post-stroke patients, cardiovascular incidents) and can be determined through time in an Up and Go test (TUG) [35,36]. In some studies, TUG was shown to be an efficient test in the prediction of falls among the elderly, as a direct relationship was observed between the occurrence of falls in the elderly and their classifications according to the test [5,37]. However, those studies also found that the mean TUG value for elderly females was statistically higher than those of the male group [5]. This difference could be explained by the aging process known as sarcopenia, which may affect the musculoskeletal system and its functional capacity and interfere with hormonal, nutritional, metabolic, and immunological factors [38]. In fact, there are differences in the physical composition of men and women throughout the lifespan that result in differences in their physical performance due to body composition [39].

According to the above discussion, and knowing that culture influences our habits and lifestyles, the aim of this study was to investigate the relationship between health state perception and health predictors among the elderly from four different countries. Four European countries with different gross domestic product (GDP) per capita (from the highest to the lowest: Italy, Spain, Portugal and Hungary [40]), different levels of physical activity in older adults (from greater to the lowest values: Spain, Portugal, Italy and Hungary [40]), and dissimilar life expectancy (from the highest to the lowest: Spain, Italy, Portugal and Hungary [40]) were included in this study. It was hypothesized that health perception is related to culture and, therefore, is different across countries and among different age-groups. It was also expected that different strength and agility performances would be obtained depending on the sex and age of participants, which could also be related to cultural differences.

## 2. Materials and Methods

### 2.1. Study Design

The present cross-sectorial study includes a questionnaire and biometric data collected from a cohort of older adults from four European Union countries (Hungary, Portugal, Italy, and Spain). All participants were healthy older adults (+60 years old) who had retired before 1 January 2018. Participants were recruited through the European Erasmus + “IN COMMON SPORTS” project, which intends to organize adapted sports competitions based on participants’ age specificities. The ethical standards of the Declaration of Helsinki were followed, and this study was approved by the Instituto Politécnico de Viana do Castelo Technical-Scientific Council (IPVC-ESDL180417).

### 2.2. Participants

The sample included 306 older adults from four countries: Hungary (HU), Italy (IT), Portugal (PT), and Spain (SP). Those countries were included as, despite being part of Europe, they have different lifestyles and socioeconomic statuses [41]. No chronic disease that hindered the practice of sports was reported by any subject. The cognitive status of participants was also classified as “mild” to “normal”, according to a mini-mental test. Their main anthropometric characteristics are expressed in Table 1. Further analyses divided the participants into three different age groups: group 1 (up to 65 years old), group 2 (65–70 years old), and group 3 (more than 70 years old). This division was performed to understand whether behavior patterns were maintained over time, verifying the effect of age in the tests covered.

### 2.3. Health-Related Quality of Life

Health-related QoL was measured using EuroQoL-5 Dimension (EQ-5D-5L) and the EQ-5D-5L index, developed by the EuroQoL group [42]. The EQ-5D-5L is a simple measure of overall health used in clinical and economic evaluations. The instrument consists of five dimensions: mobility, self-care, usual activities, pain/discomfort, and anxiety/depression. There are five response levels for each dimension: “no problems”, “mild problems”, “moderate problems”, “severe problems”, and “extreme problems”. In the present study, we combined “some problems” and “severe problems” responses into a “yes problem” response. The EQ-5D-5L index is a health-related QoL score obtained by applying a weighted value to each dimension score. Index scores range from −1 (health status worse than death) to 1 (perfect health) [43].

### 2.4. Hand-Grip Strength

HG was measured with a Jamar hand dynamometer (Sammons Preston Inc., Bolingbrook, IL, USA). The participants were asked to stand and hold the dynamometer with their arm parallel to their body without squeezing it against their body. Both hands were tested in this test. The examiner ensured that the arm to be tested was held by the participant’s side with their elbow at a 90° angle. The width of the handle was adjusted to the size of each participant’s hand to make sure that the middle phalanx rested on the inner handle. Participants were allowed to perform one test trial. After this, three trials followed, and the best score was used for analysis. HG was expressed in kilograms (Kg).

### 2.5. Timed Up and Go Test (TUG)

The TUG was developed in 1991 to examine functional mobility in the elderly [44]. This test allows for the recognition of other different diseases, mainly those related to walking activities. It has certain phases during which it is possible to obtain different readings and calculations for various features, such as sitting on a chair, lifting from the chair, walking for three meters, reverse marching, walking another three meters toward the chair, and sitting on the chair. It measures the time needed to conclude the predetermined route in seconds.

### 2.6. Statistics

Data normality was checked by applying the Shapiro–Wilk test. Descriptive statistics, such as means and standard deviations, were calculated. A three-way MANOVA was conducted to analyze participants based on their age, sex, and the country they lived in. The interaction effects in all included variables were also considered. The Bonferroni test was executed as a post hoc test. All statistical analyses were carried out using SPSS for Windows Version 22.0 (SPSS Inc., Chicago, IL, USA). *p*-values of < 0.05 were considered statistically significant.

## 3. Results

As an interaction was found, the analysis could be done by comparing the different groups that were included.

### 3.1. Age-Group Effect

As shown in Table 2, three different age groups were created: group 1 (up to 65 years old), group 2 (65–70 years old), and group 3 (more than 70 years old).

QoL was perceived as worse in the third group than in the two younger groups.

Differences were observed in HG tests for the left (F_8,740_ = 3.604, *p* < 0.05) and right hand (F_8,740_ = 6.101, *p* < 0.01). For both HG values, group 2 obtained better values than the other groups.

Regarding the TUG test, statistically significant higher values were registered by the oldest group (F_8,740_ = 7.206, *p* < 0.01).

### 3.2. Sex Effect

Table 3 expresses the differences between sexes, which were observed only for both HG values, with males presenting higher scores.

### 3.3. Country Effect

The different results for each country are presented in Table 4. The perceived QoL was higher for HU (F_12,926_ = 4.637, *p* < 0.01), IT (F_12,926_ = 4.637, *p* < 0.01) and PT (F_12,926_ = 4.637, *p* < 0.01) when compared with SP.

In the HG left, lower values were registered in HU than IT (F_12,926_ = 3.039, *p* < 0.05) and PT (F_12,926_ = 3.039, *p* < 0.01). Similarly, the results for SP were lower than for IT (F_12,926_ = 3.039, *p* < 0.05) and PT (F_12,926_ = 3.039, *p* < 0.01). Regarding HG right, PT exhibited the highest values of the four included countries. Conversely, HU subjects displayed the most robust results on the TUG test of all countries included.

## 4. Discussion

The present study analyzed the relationship between health state perception and health predictors among elderly people from four different countries. The results showed that age and country influence an individual’s QoL perception, with the oldest age group and Spanish individuals presenting the lowest scores. These results support the first hypothesis of this study. Regarding the second hypothesis, unexpected results were found, as only strength performances—determined using HG tests—were different between sexes. The agility test (TUG test) showed no differences between the sexes. Altogether, the results show that age and culture influence health state perception and health predictors among the elderly to a greater extent than sex does.

Health-related QoL is clearly influenced by several endogenous and exogenous parameters [8,9]. Among older adults, it is common to attribute health problems to “old age” rather than illness [45], which nurtures beliefs that affect health outcomes, influence health-related behaviors [46], and influence the perception of QoL. This could be one of the main justifications for the lower QoL perception of the oldest group in the present study.

Hence, a previous study that analyzed senior citizens from nursing homes and from a University of the Third Age found a relatively low correlation between physical fitness and QoL (r = 0.75 and r = 0.90, respectively), which indicates the important role of physical fitness in self-assessed QoL [10,47]. This could be another reason for the lower QoL perception in our third group (the oldest one), as these participants also performed the worst in the HG and TUG tests. Although this sample is composed of active elderly people, the results should still be considered, because other studies have shown that low expectations regarding aging were more likely to report very low levels of physical activity than those with high age-expectations, even after adjusting for sociodemographic characteristics, self-efficacy, and many indicators of health status [46].

The loss of muscle strength, which enhances the risk of fatal falls and resultant fractures, is strongly related to reduced health-related QoL [14,15]. One of the major factors contributing to the loss of strength and power is a loss of muscle mass. The age-related loss of tissue mass has been termed “sarcopenia” [48], which is different from a loss in muscle mass due to disuse atrophy or disease-induced cachexia [49]. In fact, people are vulnerable to the adverse consequences of sarcopenia and osteoporosis (i.e., frailty, increased risk of falls, disability, cognitive impairments, depression, and even mortality), especially during old age [11,12,19]. The decrease of muscle mass and muscle strength is caused by genetic factors [18], as well as by lifetime physical activity [19,20] and one’s occupation [21]. Surprisingly, in the present study, the second age group presented the highest HG values in both hands, conflicting with the expected linear inverse relationship between HG and age. This was maybe the most active and enthusiastic among the three groups, as they also tended to perform well in the TUG test.

De Paula Rodrigues et al. [37] stated that performances in TUG tests of longer than 9 s (for individuals between 60 and 69 years of age), 10.2 s (for people between 70 and 79 years of age), and 12.7 s (for people 80 to 99 years old) can be considered above average. Such values indicate that interventions are needed to reduce the risk of accidents. The results of the present work show that our sample is clearly comprised of active and healthy people, as the oldest group did even reach eight seconds. This quite easy test has been considered a useful tool for evaluating the risk of falling among the elderly [5]. It is a quick and straightforward clinical test for assessing lower extremity performance related to balance, mobility, and fall risk in the elderly population and people with pathologies [35,36,50].

It is well-known that men tend to present higher values of strength than women, both during adulthood and old age. Therefore, the higher HG values for both hands were not surprising. The low levels of endogenous testosterone largely explain the reductions in the amount and quality of muscle mass in elderly men. In elderly women, this decline has been linked to the menopause process. In fact, testosterone decreases both in men and women [51,52], although this decline does not occur at the same magnitude in both sexes [53]. When comparing sexes at the same age, Kwak et al. [54] found a link between weakened hand-grip strength and a lack of flexibility exercises in elderly men. In women, weakened hand-grip strength was related to a lack of muscle strengthening exercises. A surprising result was related to the similarity between sexes in their QoL scores, as women tend to have more complaints as bad or very poor [5]. Indeed, although women live longer than men, they report poorer health [55], as well as more physical limitations [56] and chronic conditions [57].

Considering that older people with a lower socioeconomic status (e.g., lower education level or living) were found to be more frail [58,59], it was expected that HU presented the lowest QoL scores, since they show the lowest GDP per capita [40,41], and lower life expectancy [40]. Nevertheless, the lowest QoL scores were observed in SP. In fact, it seems that physical fitness clearly influences that perception, as SP was the only country that did not stand out in either the strength or agility tests, which seems to contradict the highest values (68% in opposition to 35% in Portugal, 31% in Italy and 12% in Hungary [40]) of engagement in physical activities in older adults in that country. On the one hand, IT and PT presented greater HG left values, and PT showed the greatest HG right value among the four countries. On the other hand, HU presented the highest results on the TUG test. Physical fitness is often linked to health and is an important element in determining QoL [60,61].

Therefore, according to the HG and TUG test results, maybe the SP sample did not perform enough physical activity. Indeed, this country was the last to join the Olimpics4all project, and thus may have accumulated less physical activity compared to other countries. Other studies have shown that people with low expectations regarding aging were more likely to report very low levels of physical activity than those with high age expectations, even after adjusting for sociodemographic characteristics, self-efficacy, and many indicators of health status [46]. These findings underscore the importance of placing greater attention and investments in public health interventions with the aim of promoting physical activity participation among older adults living in the community [62].

This study has some limitations, such as the differences in sample size and sex distribution among countries. However, this population took part in a particular project that promotes sports development in the elderly. In fact, it was surprising that there were more female practitioners than male practitioners. Moreover, the team that applied the included tests was not the same in each country. However, to avoid bias in the results, a training program was carried out in advance, and data collection processes were supervised by experts in the field. Finally, in the present study, the physical activity level of the participants was not controlled. The association of physical activity level with both the HG and TUG tests is well-known. Therefore, findings should be understood with caution, and the definition of participants’ physical activity levels should be considered in future studies.

## 5. Conclusions

As hypothesized, QoL showed to be dependent on a person’s age, physical fitness, and countries. In the present study, HG and QoL values increased with age, while TUG values increased with age. The only between-sex difference was in HG, with men showing better results than women. Altogether, the results show that age and culture influence health state perception and health predictors among the elderly to a greater extent than sex does.

## Figures and Tables

**Table 1 ijerph-18-04053-t001:** Mean ± SD values of age, body mass, and height of the sample regarding age and sex for each county.

	Hungary (*n* = 86)		Italy (*n* = 133)		Portugal (*n* = 95)		Spain (*n* = 62)	
	Male (*n* = 19)	Female (*n* = 67)	Male (*n* = 29)	Female (*n* = 104)	Male (*n* = 34)	Female (*n* = 61)	Male (*n* = 11)	Female (*n* = 51)
Age (years)	69.90 ± 5.26	66.52 ± 5.50	71.80 ± 7.54	70.13 ± 7.46	71.06 ± 6.15	71.64 ± 6.68	76.28 ± 5.73	70.98 ± 7.15
Body mass (kg)	77.98 ± 13.50	75.19 ± 12.88	82.80 ± 10.05	69.88 ± 13.31	79.07 ± 9.65	67.31 ± 11.84	75.03 ± 7.91	71.28 ± 12.02
Height (cm)	171.10 ± 9.65	160.91 ± 7.39	172.58 ± 8.66	159.84 ± 6.17	168.42 ± 6.14	154.04 ± 5.98	168.64 ± 7.73	153.74 ± 5.48

**Table 2 ijerph-18-04053-t002:** Mean ± SD values of health perception (EQ-5D-5L), left (HG Left) and right (HG Right) grip strength, and the Timed Up and Go test (TUG) based on age group.

	Age-Group 1 (*n* = 79)	Age-Group 2 (*n* = 127)	Age-Group 3 (*n* = 170)
EQ-5D-5L *	0.93 ± 0.10	0.93 ± 0.09	0.88 ± 0.13
HG Left *	30.83 ± 9.18	34.92 ± 12.88	31.58 ± 11.18
HG Right *	29.27 ± 9.50	33.46 ± 12.40	29.70 ± 11.04
TUG *	6.09 ± 1.24	6.03 ± 1.25	7.40 ± 2.52

* Significant differences among ages.

**Table 3 ijerph-18-04053-t003:** Mean ± SD values of health perception (EQ-5D-5L), left (HG Left) and right (HG Right) grip strength, and the Timed Up and Go test (TUG) based on sex.

	Males (*n* = 93)	Females (*n* = 283)
EQ-5D	0.94 ± 0.09	0.89 ± 0.12
HG Left *	43.83 ± 12.36	28.84 ± 8.41
HG Right *	42.58 ± 12.39	27.04 ± 7.82
TUG	6.10 ± 1.82	6.85 ± 2.08

* Significant differences between sexes.

**Table 4 ijerph-18-04053-t004:** Mean ± SD values of health perception (EQ-5D-5L), left (HG Left) and right (HG Right) grip strength, and the Timed Up and Go test (TUG) based on country.

	Hungary (*n* = 86)	Italy (*n* = 133)	Portugal (*n* = 95)	Spain (*n* = 62)
EQ-5D *	0.94 ± 0.09	0.90 ± 0.10	0.92 ± 0.12	0.84 ± 0.14
HG Left *	29.78 ± 9.21	33.58 ± 10.82	35.57 ± 15.17	29.56 ± 7.27
HG Right *	28.80 ± 8.68	30.88 ± 10.58	34.48 ± 14.95	28.23 ± 8.18
TUG *	5.84 ± 1.15	7.16 ± 2.09	6.62 ± 2.48	6.79 ± 1.82

* Significant differences among countries.

## Data Availability

http://www.olympics4all.eu/index.php, accessed date 01 April 2021.

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
