# Peer review of "The Relationship between Health Perception and Health Predictors among the Elderly across European Countries"

_ijerph, 2021, doi:10.3390/ijerph18084053_

Round 1

Reviewer 1 Report

Interesting theme, the strongest point should be the differences in relation to the country comparison and this was low explored by the authors; since the sex and age differences are expected and well documented in the literature. It was very interesting to see the comparison of each sex and age groups by country, so it was possible to relate the studied variables differences with culture, socioeconomic, health system etc of each country. The way it is presented does not bring much novelty.

Some additional suggestions are made to improve the article quality. 

  1. Methodology: It is not clear why the authors presented HG of left and right, since considering the methodology description they just measured the dominant hand. It is important to mention the number of subjects in each hand, or just report as HG of dominant hand.
  2. In "2.5. Timed Up and Go test (TUG)" include the paper of reference and the measure unit (in this case,  seconds).
  3. It is not clear why the age groups were divided like this: up to 65 years, 65-70 years old and more than 70 years old.  The last group has a very large range. 
  4.  Table 1 sounds as results section. More details about sampling is necessary in this section.
  5. It is important to include and control by the physical activity (PA) level. The PA is strongly related to HG and TUG test. If the authors does not have any information about this must be discussed in limitations section. 
  6. Important to  update the references, less than 1/4 of the references have less than 5 years.

Author Response

The authors acknowledge the Reviewer’s comments that help us to improve the manuscript. The changes performed regarding your comments were written in red colour to facilitate your evaluation.

Regarding the specific Reviewer’s comments:

  1. In fact, we started to analyse only one hand, however, as it was observed some relationship between both hands and the other variables, we decide that those results should be included in the manuscript.
  2. Thank you for your comment. That information was added in the manuscript.
  3. That division was to understand if the age really influences the tests included. Although the groups were not equally distributed, that 5 years gap seemed more suitable to really understand that possible influence.
  4. More detailed regarding the age groups division were included in the manuscript.
  5. The PA variable was not controlled in this study. Therefore, such limitation will be added to limitations session.
  6. Thank you for your comment. An effort was done to update the references.

Reviewer 2 Report

Dear Authors

The paper addressed an important and relevant topic.

Only two comments:

  1. could provide more inofrmations regarding chronic diseases?
  2. could be analysis adjusted by BMI
  3. could be provide informations regarging cognitive status of the participants

Author Response

The authors acknowledge the Reviewer’s comments. Please find the included information regarding your comments in blue colour.

Specific comments:

  1. That information was added in the methodology section.
  2. That analysis was initially tried by the authors, however, any difference was observed in the different results. Therefore, that inclusion was not considered.
  3. As this study made part of a bigger Project, the Mini-Mental test was conducted. Regarding those values, all subject presented “mild” to “normal” results. This information was added to participants topic.

Reviewer 3 Report

The topic is interesting and I think that the article can make a valuable contribution to the scientific debate in the field of health sciences.

However, to be published in this journal, the paper should be further developed. In particular, I suggest some insights:

  • The title should better clarify the scope of the article.
  • The manuscript lacks a specific theoretical section. Please include at least a paragraph with a literature review on the topic, also highlighting the possible research gap(s) that your contribution is aiming to fill.
  • In the methodology, I would specify the reasons why you decided to focus on those four contries.
  • Conclusions should include implications for research, practice and/or society, as well as the answer to the research question stated at the end of the introduction.

Morever, I suggest you to carefully re-read the text to detect some typos and double spaces (for example, in Line 15 Hungry instead of Hungary).

Good luck!

Author Response

The authors acknowledge the Reviewer’s comments. Please find the included information regarding your comments in green colour.

Specific comments:

  1. Thank you for your suggestion. Please verify if it is clear with the new suggestion.
  2. It was added a paragraph on the introduction section trying to strengthen the basis of this study.
  3. The reason why these countries were selected were based on the fact that although they were all included in Europe, they presented different socioeconomic status.
  4. Thank you for your comment. An effort was done to include that information.

Round 2

Reviewer 1 Report

Thank you for the revised version and for considering the suggestions. 

I have two minor observations yet: 

  1. Please, review the references  6, 40 and 41, it seems to have some problems in format.
  2. In lines, 99-104 as you used the same reference it is not necessary to repeat it for 3 times, just once at the end is enough.

Reviewer 2 Report

Dear Authors

I have not further comments

Reviewer 3 Report

Dear Authors,

Thank you for the revised version of the paper. I think that the manuscript can be now accepted for publication. Congratulations!